# Gas Phase Emissions of Volatile Organic Compounds Arising from the Application of Sunscreens

**DOI:** 10.3390/ijerph20115944

**Published:** 2023-05-24

**Authors:** Amber M. Yeoman, Marvin Shaw, Martyn Ward, Lyndsay Ives, Stephen J. Andrews, Alastair C. Lewis

**Affiliations:** 1Wolfson Atmospheric Chemistry Laboratories, Department of Chemistry, University of York, York YO10 5DD, UK; li570@york.ac.uk; 2National Centre for Atmospheric Science, University of York, York YO10 5DD, UK; marvin.shaw@york.ac.uk (M.S.); martyn.ward@york.ac.uk (M.W.); stephen.andrews@york.ac.uk (S.J.A.); ally.lewis@ncas.ac.uk (A.C.L.)

**Keywords:** VOCs, mass spectrometry, personal care products, air quality, public health, contaminants

## Abstract

The speciation of volatile organic compounds (VOCs) emitted from personal care products (PCPs) is complex and contributes to poor air quality and health risks to users via the inhalation exposure pathway. Detailed VOC emission profiles were generated for 26 sunscreen products; consequently, variability was observed between products, even though they were all designed for the same purpose. Some were found to contain fragrance compounds not labelled on their ingredients list. Five contaminant VOCs were identified (benzene, toluene, ethylbenzene, o-xylene, and p-xylene); headspace sampling of an additional 18 randomly selected products indicated that ethanol originating from fossil petroleum was a potential source. The gas phase emission rates of the VOCs were quantified for 15 of the most commonly emitted species using SIFT-MS. A wide range of emission rates were observed between the products. Usage estimates were made based on the recommended dose per body surface area, for which the total mass of VOCs emitted from one full-body application dose was in the range of 1.49 × 10^3^–4.52 × 10^3^ mg and 1.35 × 10^2^–4.11 × 10^2^ mg for facial application (men aged 16+; children aged 2–4). Depending on age and sex, an estimated 9.8–30 mg of ethanol is inhaled from one facial application of sunscreen.

## 1. Introduction

UVA and UVB radiation from the sun are not filtered by the Earth’s atmosphere and can cause (or exacerbate) sunburns, ageing linked to long-term skin damage, and skin cancer—both non-melanoma, one of the most common cancers in the world, and the less common melanoma [1,2]. Suitable protection has been sought for millennia, with the ancient Egyptians and Greeks employing natural remedies such as flower extracts and olive oil to protect their skin [3]. Commercial sunscreen products are formulated with active UV-filtering ingredients that prevent radiation from reaching, and thus damaging, the skin. Organic or chemical filters, with the latter including compounds such as octocrylene, oxybenzone, and homosalate (among others), function by absorbing UV radiation. Physical filters, also known as inorganic or mineral filters, function by reflecting UV from the skin (although there is research suggesting that they also protect skin through absorption) [4]. The most common physical filters are zinc oxide and titanium dioxide.

Whilst a range of UV-filtering options are available, consumer demand has led to a growth in different formats for delivering sunscreens. They exist as creams and lotions, liquid sprays, aerosols, oils, mists, balms, gels, and powders, and some claim additional moisturising, anti-ageing, water-resistant, and single-application properties. Some are formulated for more sensitive skin and are usually alcohol- and/or fragrance-free; these are often marketed for use on children or for specific use on the face. Less commonly, products are specially formulated for use on the scalp and hair [5].

As sunscreen products serve a more critical preventive pharmaceutical purpose than many other personal care products (PCPs), there are tighter rules related to their formulation and sale. They are regulated and distributed as over-the-counter drugs in Canada [6] and the US [7], while they are only classed as cosmetics in the EU [8]. There are guidelines surrounding labelling, performance claims, and efficacy; lists of approved UV filters; and safety regulations to ensure that all ingredients, not just the active ones, are safe for use. Additionally, sunscreen products display a sun protection factor (SPF) rating, which is defined as “the ratio of UV energy needed to produce a minimal erythemal dose on protected to unprotected skin [9,10]”, to ensure the consumer receives correct information with which to make informed choices regarding the protection of their skin.

As is the case for most PCPs, the key safety concerns for sunscreens are adverse skin reactions, dermal absorption, and endocrine effects [9,11], all of which are related to dermal exposure pathways. Yeoman et al., (2021) [12] demonstrated how the inhalation exposure pathway can also be important since PCPs contain, and thus emit, a wide range of volatile and semi- volatile organic compounds (VOCs and SVOCs). There is currently limited information on the types and amounts of VOCs emitted from sunscreens, even though facial application is perhaps the most common end-use. Garrido et al., (2019) [13] published an exposure assessment for the air-to-skin uptake of semi-volatile organic compounds (SVOCs) from sunscreens that focused on the dermal exposure route. Research suggests that it is possible for titanium dioxide and zinc oxide particles to be inhaled from mineral sunscreens, particularly when used in spray or powder formulations [14,15], but no estimates have been made regarding the potential inhalation of VOCs from sunscreen products.

Recently, the California Air Resources Board (CARB) proposed amendments to their Consumer Products Regulations, identifying aerosol sunscreens as a product category that should be reviewed [16]. This was likely because aerosol propellants are known to contribute substantially to overall anthropogenic VOC emissions and poor outdoor air quality [17,18], with many other aerosolised products also being selected for evaluation. PCPs emit many other VOC species alongside propellants (most commonly i- and n-butane and propane). PCP emissions can include fragrances and solvents [19,20,21,22]; thus, it is likely that non-aerosolised sunscreens also contribute modestly to emissions related to outdoor air quality. Fragrance compounds, such as monoterpenes, can induce adverse effects when inhaled in high concentrations [23] and have the potential to react with ozone and hydroxyl radials to form secondary organic aerosols (SOAs) [24] and small carbonyls such as formaldehyde. Other common PCP solvents such as ethanol also react with hydroxyl radicals [25]. Sunscreens have a higher probability of being applied outdoors than other classes of PCPs, making them potentially less impactful on indoor air quality but, through personal exposure, still potentially important if facially applied.

Unlike many classes of PCPs, sunscreens are used by a wider cross-section of ages and are one of only a small number of PCPs regularly used on children and infants. Children are potentially more vulnerable to the effects of air pollutants as they inhale more air relative to their body weight than adults and their respiratory systems are less developed [26]. Research has linked the exposure of children to domestic VOCs to the diagnosis of asthma in those aged between 6 months and 3 years old [27], other respiratory diseases affecting those under 36 months of age [28], and the exacerbation or induction of multiple allergic symptoms, rhinitis, and eczema, among other ailments [29].

The aim of this study is to evaluate the variability and speciation of VOCs emitted from a range of sunscreen products, covering different application types, SPF ratings, and intended users. The product-specific emission rates of VOCs are then compared to other PCPs and similar emission sources. An unanticipated consequence of this study has been the identification of potential trace contaminant aromatic VOCs present in a small number of sunscreen products.

## 2. Materials and Methods

### 2.1. Sunscreen Selection Process

Twenty-six sunscreen products from fifteen different brands were purchased from a major UK supermarket retailer. They were selected so that a range of different formulations and SPF ratings could be represented (Figure 1), including some products that were marketed specifically for use on the face (12 products) or for children (4 products). As creams and lotions are the most used forms of sunscreen, our selection was a proportionate representation of what is available in the UK market and, by extension, what consumers realistically use. There were no aerosolised (incl. mists), roll-on, stick, or powder sunscreens studied, as the methodology was designed for liquid samples. It is likely that some of the products in this latter group, particularly stick and powder sunscreens, would have lower VOCs emissions given that they do not contain a solvent matrix.

### 2.2. VOC Screening and Initial Identification Using GC/MS QTOF

An Agilent Technologies (Santa Clara, CA, USA) 7200 Accurate-Mass Quadrupole-Time-of-Flight Gas Chromatography/Mass Spectrometer (Q-TOF GC/MS) fitted with a Gerstel (Gerstel, Germany) MultiPurpose Autosampler and a 2.5 mL headspace syringe was used for qualitative screening and identification of VOCs present in the headspace of each of the 26 products.

Around 2 g of sunscreen product was weighed in 20 mL clear glass, round-bottom headspace vials and capped with magnetic stainless-steel screw caps with 1.6 mm PTFE-faced butyl septum.

Separation was achieved using a BPX5 50 m × 0.32 ID gauge 1 µg capillary column with a flow rate of 2 mL min^−1^ using helium gas. An N2 collision gas was used at a flowrate of 1.5 mL min^−1^ in the mass spectrometer, collecting data at an effective frequency of 5 Hz. The GC headspace needle was maintained at 70 °C. Samples were incubated and agitated for 5 min in a Gerstel agitator, once at 35 °C and once at 50 °C (2 samples run per product), before a 2 mL headspace sample was taken. The GC inlet temperature was set at 330 °C with a split ratio of 10:1 and split flow of 20 mL min^−1^. The column oven temperature was initially 40 °C, which was held for 5 min and then increased to 340 °C at a rate of 10 °C min^−1^, amounting to a run time of 40 min. The transfer line was set to 350 °C. Data were acquired over a mass range of 28–500 amu.

### 2.3. Identifying VOC Contaminants

Eighteen random PCPs from two UK retailers were selected for analysis (Classifications in Appendix A). Nine were specifically selected due to their content of ‘organic’ or ‘organically sourced’. The other nine products were assumed to have ethanol derived from fossil petroleum. The same GC/MS QTOF preparatory procedure and methodology were used for the sunscreen products, with the agitator set at 50 °C.

### 2.4. Emission Rate Measurements Using SIFT-MS

A Voice200 SIFT-MS was used in SIM (selected ion monitoring) scan mode to quantify emission rates of 15 different VOC species from each of the sunscreens. These VOCs were chosen based on the results from the earlier GC/MS QTOF screening and represented the most abundant and commonly found compounds. The emissions estimates were made following a methodology previously reported in the work of Yeoman et al., (2020) [19], with the SIFT-MS instrument sampling from the headspace of a gas-tight sample vessel where the sample was placed. Targeted VOCs and the *m*/*z* values used to identify them are listed in Appendix A.

Approximately 2 mg of each product was weighed in a plastic vial cap and placed into a 50 mL stainless-steel gas-tight sample vessel, which, in turn, was thermostatted at 35 °C for 1 h to simulate skin temperature. Driving off VOCs at a higher temperature would not be representative of emissions present during normal product use. A gas phase headspace sample was drawn continuously from the gas-tight vessel into the SIFT-MS at a flowrate of 10 mL min^−1^ under atmospheric pressure, with the inlet to the vessel connected to a supply of high-purity N2. The sampling lines, vessel, and instrument were operated without sample for 5 min prior to the product being introduced, with this initial period used as a baseline measurement, which was later subtracted from the sample measurement data.

Data acquisition lasted for 65 min, with an ion dwell time of 100 ms per *m*/*z* and a cycle time per reagent ion mass spectra of 38 s, amounting to 114 s overall. Over the 60 min analysis period, this process provided an average 32 mass spectra per reagent ion.

### 2.5. SIFT-MS Calibration

SIFT-MS calibration was performed using two separate methods. The first was an in-house dynamic liquid calibration system, as described in the study by Yeoman et al., (2021) [12]. The second involved an in-house dynamic gas-blending calibration system, as described in the study by Wagner et al., (2021) [30]. Both techniques are briefly described in the following paragraph.

For the dynamic liquid-based calibration system, an aqueous solution of water-soluble t-butyl alcohol was created in deionised water with a target mixing ratio of 1000 ppb. An SIFT-MS SIM scan was run as previously described since the H_2_O liquid flowrate was changed by increments of 0.1 H_2_O g h^−1^, starting at 0.1 H_2_O g h^−1^ and finishing at 1.3 H_2_O g h^−1^, thus providing 13 calibration points. Similarly, a solution of D5 was prepared in cyclopentane with a target mixing ratio of 650 ppb, and the ‘H_2_O’ liquid flowrate was changed from 0.5–2.0 H_2_O g h^−1^ by increments of 0.5 H_2_O g h^−1^, providing 4 calibration points. These solution flowrates were nebulised and subsequently de-solvated into nitrogen (BOC, research-grade), which was executed at a flowrate of 1000 sccm. Obtained concentrations were in the 0.2–1.7 and 2–7 ppm range for t-butyl alcohol and D5, respectively. Previously calculated liquid calibration correction factors for ethanol, 2-propanol, and benzyl alcohol were applied to the data [12]. Liquid calibration curves can be found in Appendix A. For the calibration of limonene, benzene, and toluene, a VOC standard (1 ppm in nitrogen certified by the National Physical Laboratory, UK) containing these compounds was dynamically diluted into a zero air flow of 1000 sccm (generated from a palladium-based VOC scrubber heated to 380 °C), which allowed for controlled dilutions in the ppb range.

### 2.6. Data Workup and Analysis

GC/MS Q-TOF data were analysed using Agilent MassHunter Qualitative Analysis (version B.07.00) and the latest version of the NIST Library (MS Search v2.3) to identify compounds.

The presence of several compounds was confirmed by employing authentic standard materials using the same method as previously employed (agitator 35 °C) but with minor changes to the conditions (Table 1). Those compounds developed in ethanol were 1 mL analyte in 25 mL volumetric.

Additional standards for pentane, cyclohexane, hexane, heptane, and diethyl ether were tested in order to identify contaminants in the random organic/non-organic ethanol PCPs with a 10:1 split (20 mL min^−1^). All compounds except pentane and diethyl ether were prepared in ethanol with a 4 min solvent delay. All primary SIFT-MS data workups were carried out using the SIFT-MS instrument running LabSyft software.

### 2.7. Calculating Usage Dose

For sunscreen use calculations, body surface area (BSA) was calculated using Equation (1), i.e., the Du Bois formula.
(1)$$BSA=0.007184\times weight^{0.425}\times height^{0.725}$$

National Health Service data from 2018 concerning average height and weight according to age and sex are detailed in Table 2 in the first two columns [31]. Equation (1) was applied to these data and presented in Column 3. Taylor and Diffey (2002) [32] describe 11 areas of the body to which 2 mg cm^−2^ sunscreen should be applied. Each area represents 9% of a person’s total BSA, amounting to 99% in total, as shown in column 4. Recommended sunscreen use in grams has been calculated using 99% BSA and is presented in column 5. The same method for calculating head, neck, and face application is applied and presented in columns 6 and 7.

Emission-to-inhalation ratios, representing the fraction of VOC inhaled relative to the amount emitted overall to the gas phase in air with respect to ethanol (0.72), t-butyl alcohol (0.16), 2-propanol (0.11), benzyl alcohol (0.44), and limonene (0.64), were calculated based on current data from Yeoman et al., (2021) [12]. These ratios were applied to the corresponding compounds to estimate inhalation from head, neck, and face application. The average of these ratios was taken (0.3588) and applied to all the remaining VOCs.

## 3. Results and Discussion

### 3.1. Qualitative Analysis—Determination of Volatile Species Using Q-TOF GC/MS

#### 3.1.1. General Observations

Initial Q-TOF GC/MS qualitative analysis revealed substantial variability in the VOC composition of the individual sunscreens, although a relatively modest subset of around 40 different species captured the totality of all the products tested (see Figure 2). Most VOCs were identified individually; however, terpene fragrance compounds (often a complex mixture in its own right) were grouped together, and dimethicone (also known as polydimethylsiloxane) was reported as one compound. These ingredient VOCs were then further categorised into six functional categories: solvents, UV blockers, contaminants, miscellaneous, siloxanes/silanols, and fragrances. What is immediately clear is that there is significant variation in the formulations of the VOCs, even though all the products were manufactured with/for the same end-purpose.

#### 3.1.2. Solvent VOCs

It would be expected that simple solvents would be found in this kind of consumer product, particularly ethanol and its denaturants t-butyl alcohol, propyl alcohol, and 2-propanol, which are collectively known and labelled as denatured alcohol. Denatured alcohol is widely used to solvate and aid the delivery of active ingredients as well as mattify the skin and support a product’s ability to dry quickly [12].

Additionally, we found a range of other volatile solvent compounds in the products tested, which had not been detected in previous studies. Diisopropyl adipate is a popular solvent in sunscreen products as it lightens the texture of the product from heavy oil-soluble chemical UV blockers and is also an emollient and conditioner. Dipropylene glycol (DPG) has both solvent- and fragrance carrier/-fixing properties, concerning the binding and carrying of fragrance components. Fragrance fixatives help fragrance compounds disperse more slowly, increasing their volatisation time and thus lengthening the fragrance life of the product [33]. DPG is often added to perfumers’ alcohol, which is a solution of denatured alcohol and a co-solvent and a fixative. Dibutyl adipate is also a fragrance solvent but was not found to be volatile at 35 °C in any of the products tested. Therefore, we assume that it poses a lower risk of volatisation at normal usage temperatures.

#### 3.1.3. UV Blocker Compounds

As of 2022, the European Commission has approved 28 chemical UV-blocking compounds (ANNEX VI of the Cosmetics Regulation (EC) No. 1223/2009) [34]. In this study, two of them were observed to be volatile at plausible usage temperatures: homosalate (HMS) and 2-ethylhexyl salicylate (EHS). As is common, there are very few toxicology data on inhalation risks, and there is a lack of research on the atmospheric fate of either of these compounds. There are differing opinions on inhalation toxicity; the French agency for food, environment, and occupational health and safety stated that due to the lack of data on the inhalation route and low acute toxicity for the oral and dermal routes, HMS was of no concern for acute inhalation toxicity [35]. There are no UK workplace exposure limits for either compound, suggesting similar conclusions. Other sources, however, consider both UVBs to be toxic indoor VOCs pollutants. A risk assessment by Bayati et al., (2021) suggested that HMS produced adverse health effects at low exposure levels [36]. Additionally, HMS and EHS were two of the five most toxic compounds identified in the cited indoor air study.

Not only are there more human safety regulations for UV blocker compounds than other PCP solvent ingredients but there is also more public interest in their wider ‘environmental-friendliness’, which was possibly triggered by the discovery that some UV blockers damage coral reefs [37,38,39] and threaten other aquatic environments such as wastewater, lakes, and rivers [40,41,42].

The EcoSun Pass tool [43] is a method used to evaluate all potential environmental impacts of UV blockers, scoring them on biodegradation, bioaccumulation, acute aquatic toxicity, chronic aquatic toxicity, sediment toxicity, and chronic terrestrial toxicity. Atmospheric reactions and photodegradation by-products are not considered possible environmental hazards of sunscreens, which is a common assumption made about consumer products when the end-product ends up airborne rather than in aquatic systems. Little is known about the potential atmospheric fate of either HMS or EHS beyond photodegradation data regarding their reaction with OH radicals, with half-lives of 0.25 days and 0.49 days, respectively [44]. Details on potential reactions with other species in the atmosphere have not been well documented.

#### 3.1.4. Contaminant VOCs Found in Sunscreens

We identified five monoaromatic VOCs in certain sunscreen products that we contend should be classed as unintentional contaminants, as we do not believe they have been added to the products as performance-enhancing ingredients. These VOCs are benzene, toluene, ethylbenzene, o-xylene, and p-xylene (often collectively referred to as BTEX).

Benzene is a known carcinogen that the World Health Organisation’s (WHO) indoor air quality guidelines have identified as one of nine commonly present indoor pollutants found in concentrations that may be damaging to health [45]. These guidelines note the links between exposure to benzene and acute myeloid leukaemia and how concentrations of less than 1 ppm can be haematoxic. The other contaminant species have associated negative health impacts from inhalation exposure, although they are not as frequently cited as being significant indoor pollutants in terms of health impacts [46,47,48].

A potential source of contaminant VOCs in sunscreens is bulk ethanol [49], with the benzene being an unintended residue from either the purification process via distillation or from its original source. Ethanol used in consumer products can be sourced from either petroleum-derived products via the hydrolysis of ethylene or from the fermentation of organic materials such as corn, maize, sugarcane, rye, grape, and wheat crops [50,51]. Crude oil is known to contain BTEX, cyclohexane, and other hydrocarbons [52], and these may be present in trace amounts in the ethanol that is derived from it. Figure 3 shows the results of a small study on the relationship between the contaminants present in the products and their ethanol source (presumed to be petroleum-derived or labelled as organically sourced). It is assumed that organically derived ethanol has not been sourced from a petrochemical feedstock, although faith must then be placed in the veracity of the manufacturer’s labelling and supply chain.

The contaminant species identified are presented in Figure 3.

The purification process of ethanol is an additional potential source of contaminants. Distillation is often used to increase the purity of ethanol; it has a purity ceiling of ~96% as ethanol and water form an azeotropic mixture. To overcome this limitation, a dehydration process (heterogeneous azeotropic distillation) is often used, whereby an entrainer is added, such as a small quantity of benzene, n-pentane, cyclohexane, hexane, n-heptane, isooctane, acetone, or diethyl ether [53], to produce a ternary azeotrope. The ternary azeotrope has a lower boiling point, and is thus more volatile, than the ethanol–water azeotrope [54]; therefore, more ethanol can be fractionally distilled, resulting in 99.9% alcohol by volume. The result is anhydrous ethanol, which may contain several ppm of contaminant entrainer [55,56]. Both anhydrous and azeotropic ethanol are used in cosmetics. Anhydrous alcohol can be produced from fermented ethanol and still be considered ‘organic’, as 99.9% organic ethanol is available for purchase for use in PCPs. Therefore, if this is the source of contaminants, it is also likely present in organically sourced ethanol. Other feasible sources of contaminants include leaching from product packaging and the containers/equipment used during manufacturing.

An additional observation is the presence of acetone in over half of the organic products and its absence in the presumed petroleum-derived products. There does not appear to be an ‘organic’ or ‘natural’ way to easily produce acetone, so it is unclear why it is more prevalent in organic products. We note that the sample size in this study was small; thus, we must be cautious when drawing definitive conclusions.

#### 3.1.5. Miscellaneous

PCPs contain VOCs with unusual functionalities that would not necessarily be considered in traditional atmospheric chemical science or indeed as being present in ambient air as air pollutants. These include, for example, perfume fixatives and preservatives (such as 2-phenoxyethanol) and absorption enhancers (e.g., isopropyl myristate). Perhaps unsurprisingly, their inhalation toxicology and atmospheric fate in indoor environments and outdoors are unknown.

#### 3.1.6. Siloxanes/Silanols

Siloxanes and silanols are often used in PCPs to improve a product’s texture and are popular with manufacturers due to their stability and safety on the skin. The most commonly used members of this group are cyclic siloxanes (D4 and D5, also known as cyclomethicone), dimethicone, and dimethiconol. Our analysis showed that what is often labelled as simply ‘dimethicone’ is actually made up of several different (C_2_H_6_OSi) moieties even though it is labelled as one compound, which is likely carried out for convenience/brevity. Those compounds found were octamethyltrisiloxane (n = 1), decamethyltetrasiloxane (n = 2), dodecamethylpentasiloxane (n = 3), tetradecamethylhexasiloxane (n = 4), and hexadecamethylheptasiloxane (n = 5), which we refer to as Di1, Di2, Di3, Di4, and Di5, respectively.

#### 3.1.7. Fragrances

The presence of fragrance compounds in consumer products, and as volatile emissions and possible irritants, has been comprehensively discussed [20,23,57,58,59,60,61] in the literature. Typically, fragrant VOCs are not listed individually on personal care products due to a combination of issues including an excessive amount of information, complexity, and intellectual property concerns. Instead, a catch-all term such as parfum or fragrance is often included on labels.

In this study, eight sunscreen products had no parfum/fragrance or individual fragrance compounds listed in their ingredients, which are denoted by a * in Figure 4, of which three were explicitly designed for use on children or infants (S3, S6, and S15). Despite their labels, five of the eight products were found to contain at least one fragrance compound, including S6, which is designed for children.

The fourth product designed for children, S21, was more highly fragranced compared to many of the other products. Studies on the effects of the inhalation of fragrance VOCs on infants and toddlers is lacking, but children may be more sensitive to inhalation exposure than adults, presenting an increased probability of developing contact dermatitis on underdeveloped skin [62].

### 3.2. Quantitative Analysis—Emission Rates of Prevalent Species

Using the data obtained from the products screened using Q-TOF GC/MS, a subsequent targeted emissions rate evaluation was undertaken using SIFT-MS. Following the method reported in the study by Yeoman et al., 2020 [19], the VOC emissions rate from each product was quantified in the units of micrograms of VOC emitted per second per gram of product used. Across the range of products, which were all designed for the same application and purpose, there was considerable variability in the emission rates of common compounds, as shown in Figure 5.

#### 3.2.1. Usage Dose

Table 3 and Table 4 present the results of applying the recommended sunscreen doses calculated in Table 2 according to BSA to the mean values presented in Figure 5.

The amount of each compound inhaled after facial use has been estimated by applying headspace/inhaled ratios to the headspace data in Table 4, and the results are presented in Figure 6 (Appendix A). According to our estimates, it is plausible that a toddler (a child aged 2–4 years) might be inhaling up to 10 mg of ethanol from one facial application of sunscreen, while adults (age 16+) may be exposed to 26–30 mg via the inhalation pathway. The only public health guidelines in the UK regarding inhalational exposure to controlled substances (of which benzene, ethanol, ethyl acetate, ethylbenzene, 2-propanol, t-butyl alcohol, and toluene are considered members) are workplace exposure limits [63], which “are set in order to help protect the health of workers” and, therefore, only concern adults. The total estimated amount inhaled is equal to 4.91% of the total weight of the product and 39% of the total VOCs available.

Data on the application and inhalation of VOCs contained in facial moisturisers are only recently emerging, and methods are being constantly refined. In this study, we used our current best estimates of the fraction of VOCs inhaled from an applied facial dose, updating the estimates made in the study by Yeoman et al., 2021 [12]. The mean inhaled ethanol dose from one application of facial moisturiser (0.45 g) is estimated to be 0.29 mg, which is a factor of 10 lower than that estimated for the facial application of sunscreen. This finding is a result of more sunscreen product being used in an average application as opposed to sunscreens containing more ethanol. The mean inhaled limonene dose for facial moisturisers (1.3 × 10^−2^ mg) is approximately the same as that which we estimated for sunscreens.

Zhou et al., (2017) [64] estimated toddlers’ and infants’ inhalation of fragrance compounds in baby bath PCPs after replicating their use and measuring air concentrations. This work is one of the only known studies to consider child inhalation exposure through the direct application of PCPs. This work measured air concentrations of 1–5 µg m^−3^ for seven common fragrance ingredients (isoamyl salicylate, amyl salicylate, benzyl acetate, dimethylheptenal, methyl benzoate, α-ionone, and β-ionone) in an infant’s breathing zone. More specifically, the conducted baby lotion application experiments showed that the samples collected closest to the inhalation exposure pathway (7.6 cm from the application site) had higher concentrations than those samples located further away. These data were converted to µg by accounting for room size (16.8 m^3^) in order to directly compare them with our data (Appendix A), as follows: isoamyl salicylate 24.9 µg, amyl salicylate 23.5 µg, benzyl acetate 68.2 µg, dimethyl heptenal 21.8 µg, methyl benzoate 34.3 µg, α -ionone 27.7 µg, and β -ionone 28.4 µg. Although only one common fragrance compound was measured, the datapoints are of the same magnitude; we estimate a level of toddler inhalational exposure to benzyl acetate of 87 µg, which can be compared to Zhou’s 68 µg. This is also true for limonene (78 µg), which we used to represent all monoterpene fragrance compounds.

#### 3.2.2. Concentration of Contaminants

The concentration at which a contaminant species is present in a product is crucial for determining whether it might be of concern. According to the IUPAC definition, a trace element has “an average concentration of less than about 100 ppm atoms, or less than 100 mg kg^−1^ [65]”. Benzene, toluene, and ethylbenzene are prohibited substances and as such have no regulatory concentration limits. Under Article 17 of EU regulation No 1223/2009, traces of prohibited substances are permitted if the impurities stem from the manufacturing process, storage, or migration from packaging, which are processes that are technically unavoidable when maintain good manufacturing practices [34]. Evidence of this technical unavoidability must be provided along with precise definitions of the impurities and their sources (in relation to the potential release from packaging) [66]. Additionally, a safety assessor must decide whether the levels of the prohibited trace substances are toxicologically acceptable and whether the product is safe.

The Health-and-Safety-Executive (HSE)-based workplace exposure limits for short-term exposure (15 min) [63] for the three contaminants measured via SIFT-MS are as follows: toluene—100 ppm/384 mg m^−3^ and ethylbenzene—1 ppm/5.52 mg m^−3^, while no short-term limit is given for benzene. None of the measured mixing ratios of contaminants of the 26 sunscreens tested (Appendix A) reached these regulatory limits, as presented in Figure 7.

In May 2021 in the US, it was found that several sunscreen and sun-care products were contaminated with benzene [67]. Any product that was found to have 0.1 ppm or higher of benzene in the headspace gas phase was considered to have ‘significantly detected benzene’ and was and subsequently recommended for recall. This is considerably lower than HSE exposure limits, with the report commenting overall that “any significant detection of benzene should be deemed unacceptable” in these products. The measurement technique we used in this study is very sensitive and is able to detect mixing ratios well below the 0.1 ppm guideline. After calculating the average percentage of contaminant per total product weight, we found toluene constitutes 0.0119%, benzene constitutes 0.0006%, and ethylbenzene constitutes 0.0018%.

## 4. Conclusions

Combining the techniques of QTOF-GC/MS and SIFT-MS, we identified the VOCs off-gassed from a range of liquid sunscreen products and determined emission factors per gram of product. Our experiments show that a wide range of different VOC emissions are released from this product class both in terms of speciation and the amount released into the gas phase. Estimates were made regarding the likely inhaled dose arising from the facial application of sunscreen for adults and children. We note that the number of VOCs inhaled from sunscreen is higher than that from facial moisturisers, which is potentially because larger amounts are used in each application. The effects of inhaling VOCs from personal care products are scarcely considered in safety assessments, but this work provides a guide to a user’s likely exposure. These conclusions could be further consolidated by replicating real-life use and directly measuring inhalation using an inhalation replica. We identify that small amounts of benzene and other contaminant aromatic compounds are found in some products that contain ethanol of a fossil origin. Organic ethanol-based products had a higher likelihood of containing acetone, which is also likely a contaminant. In all cases, however, the absolute amounts of these species were low (ppb amounts) relative to the published safety standards. It should be noted that as the sample size for this investigation was small, the presence of contaminant species needs to be explored further in a separate, larger study. Some products that were purportedly fragrance-free according to their ingredients lists did, in fact, contain them; in general, we found that sunscreen products are highly fragranced. Inhalational exposure to the VOCs in these products poses unknown toxicological impacts, but we do note their common usage on children in contrast to many other PCPs that are primarily used by adults.

## Figures and Tables

**Figure 1 ijerph-20-05944-f001:**
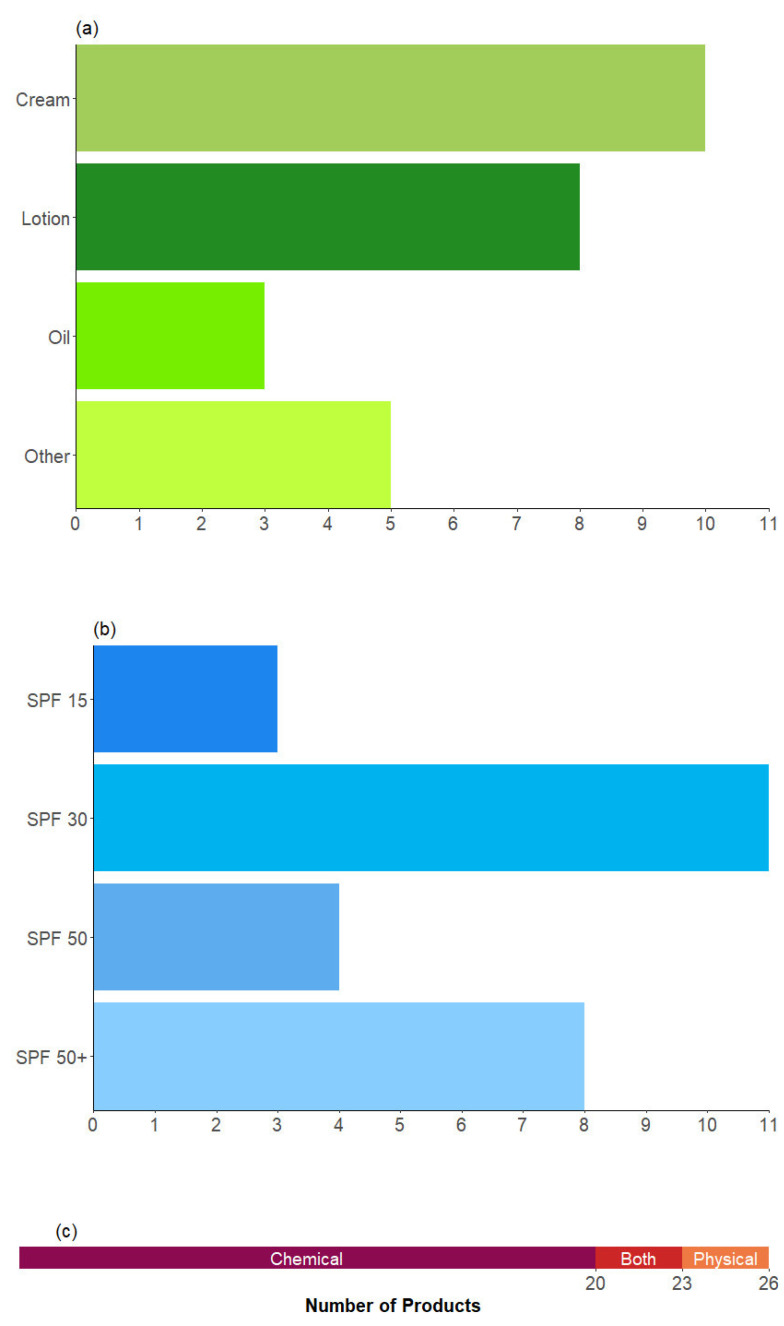
Number of products tested, which were categorised by (**a**) formulation type, with ‘other’ representing miscellaneous formulations; (**b**) SPF rating; and (**c**) UV-filtering method.

**Figure 2 ijerph-20-05944-f002:**
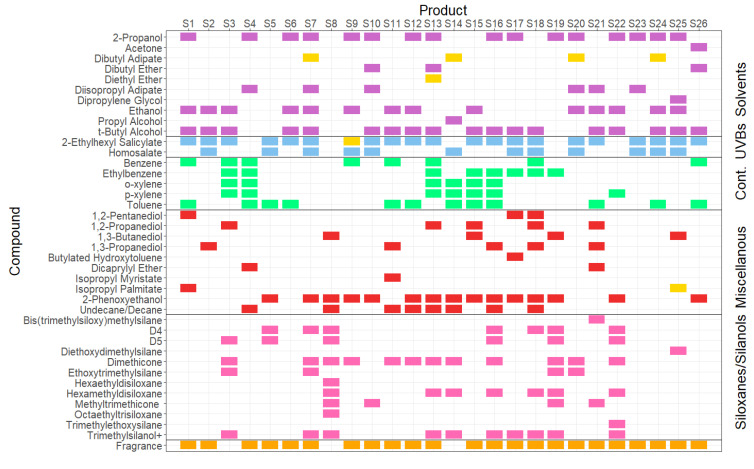
Visualisation of VOC speciation in 26 different liquid-based sunscreen products available in the UK. Yellow markers indicate solely the VOCs that were found when the sample was heated to 50 °C. UVBs—UV Blocker compounds. Cont. = Contaminants (likely unintentionally present).

**Figure 3 ijerph-20-05944-f003:**
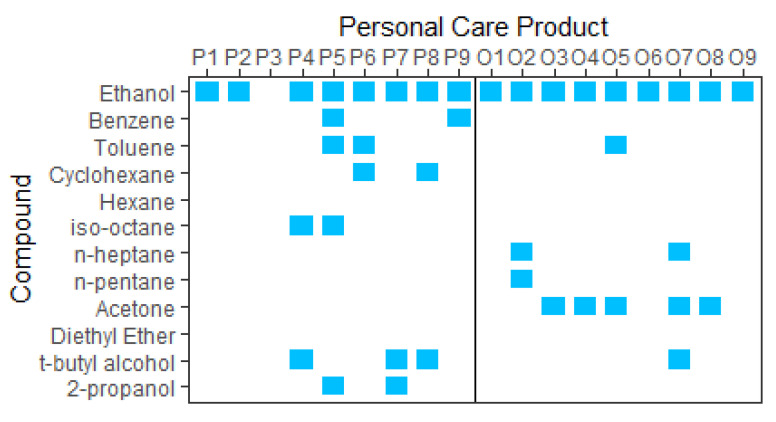
P = petroleum-derived; O = organic. Denaturants t-butyl alcohol and 2-propanol are included for comparison and analysis.

**Figure 4 ijerph-20-05944-f004:**
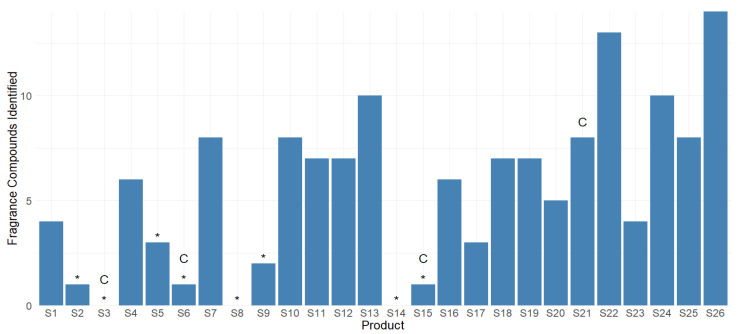
Number of individual fragrance VOCs found in each product using headspace analysis and GC-TOF/MS. * = Complexity of the parfum/fragrance or fragrance compounds not listed individually in ingredients or in products that claim to be fragrance-free. C = Formulated specifically for use on children.

**Figure 5 ijerph-20-05944-f005:**
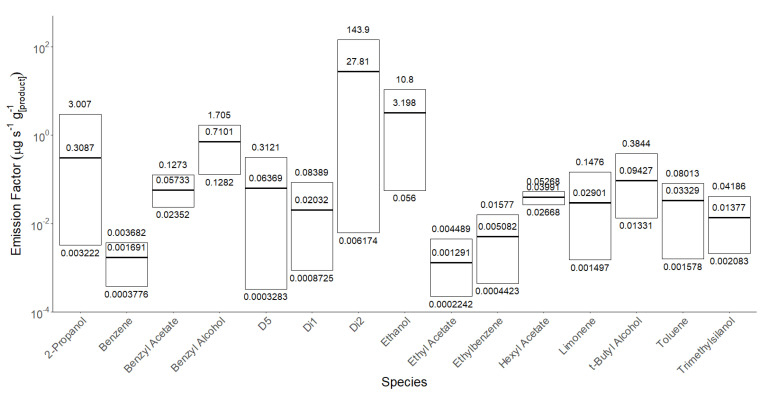
Emission factors for 15 of the most commonly found VOCs in 26 different sunscreens thermostatted at 35 °C, showing max, min, and mean emissions. Products for which QTOF results showed that a particular VOC was not present in the original formulation were excluded from the statistics.

**Figure 6 ijerph-20-05944-f006:**
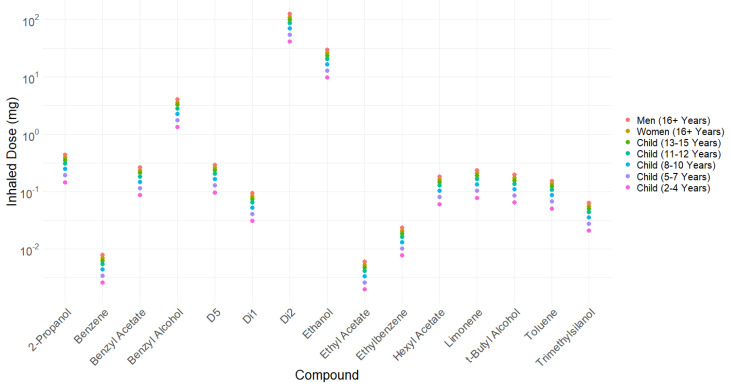
Inhalation estimates for individual VOCs based on data on a single facial application of sunscreen (presented in Table 4) and headspace/inhaled ratios. Dataset can be found in Appendix A.

**Figure 7 ijerph-20-05944-f007:**
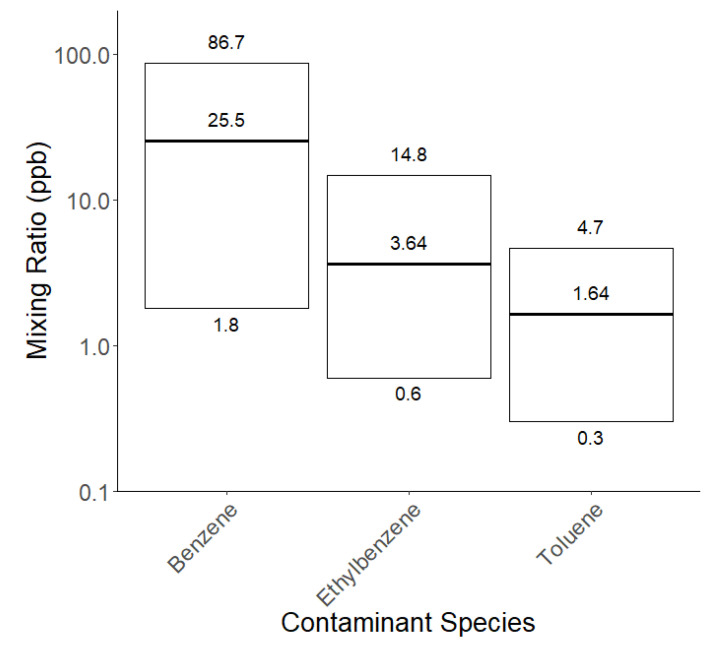
Gas phase mixing ratios of contaminant species off-gassed from sunscreens for each of the 26 sunscreen products.

**Table 1 ijerph-20-05944-t001:** Q-TOF GC/MS conditions for standards used.

Compound	Made Up in Ethanol	Solvent Delay (4 min)	Gas Saver on (15 mL min^−1^ after 2 min)	Split
1,3-Propanediol				10:1 (20 mL min^−1^)
2-Propanol				10:1 (20 mL min^−1^)
Acetone				10:1 (20 mL min^−1^)
Benzene	Y	Y	Y	100:1 (200 mL min^−1^)
Benzyl Alcohol				10:1 (20 mL min^−1^)
Di2 *	Y	Y	Y	10:1 (20 mL min^−1^)
Di3 **	Y	Y	Y	10:1 (20 mL min^−1^)
Ethanol				10:1 (20 mL min^−1^)
Ethyl Acetate				10:1 (20 mL min^−1^)
Ethylbenzene	Y	Y		20:1 (40 mL min^−1^)
o-Xylene				10:1 (20 mL min^−1^)
Propyl Alcohol				10:1 (20 mL min^−1^)
p-Xylene	Y	Y		20:1 (40 mL min^−1^)
t-Butyl Alcohol				10:1 (20 mL min^−1^)
Toluene	Y	Y	Y	10:1 (20 mL min^−1^)
Trimethylsilanol	Y	Y		20:1 (40 mL min^−1^)

Y = yes. * Decamethyltetrasiloxane. ** Dodecamethylpentasiloxane.

**Table 2 ijerph-20-05944-t002:** Height; weight; calculated BSA; full body area to which sunscreen is applied (99% BSA); head-, neck-, and face-only areas to which sunscreen is applied (9%); and recommended use in grams.

Sex and Age	Height (cm)	Weight (kg)	BSA (cm^2^)	99% BSA (cm^2^)	Recommended Sunscreen Use—Full Body (g)	9% BSA (cm^2^) For Head, Neck, and Face	Recommended Sunscreen Use—Head, Neck, and Face (g)
Men (16+ years)	175.6	84.8	2.01 × 10^4^	1.99 × 10^4^	39.8	1.81 × 10^3^	3.62
Women (16+ years)	162.1	72.4	1.77 × 10^4^	1.76 × 10^4^	35.1	1.60 × 10^3^	3.19
Child (13–15 years)	163.8	56.8	1.61 × 10^4^	1.60 × 10^4^	31.9	1.45 × 10^3^	2.90
Child (11–12 years)	152.7	46.1	1.40 × 10^4^	1.39 × 10^4^	27.8	1.26 × 10^3^	2.52
Child (8–10 years)	136.5	33.5	1.13 × 10^4^	1.12 × 10^4^	22.3	1.02 × 10^3^	2.03
Child (5–7 years)	119.3	23.4	8.79 × 10^3^	8.70 × 10^3^	17.4	7.91 × 10^2^	1.58
Child (2–4 years)	100.0	16.1	6.60 × 10^3^	6.53 × 10^3^	13.1	5.94 × 10^2^	1.19

**Table 3 ijerph-20-05944-t003:** Estimated VOC emission dose from standard full-body application of sunscreen based on age and sex (Table 2) calculated from mean emission factors in Figure 5 (Appendix A).

Species	Full-Body Application Based on Age and Sex (mg)						
	Men (16+ Years)	Women (16+ Years)	Child (13–15 Years)	Child (11–12 Years)	Child (8–10 Years)	Child (5–7 Years)	Child (2–4 Years)
Benzene	2.41 × 10^−1^	2.12 × 10^−1^	1.93 × 10^−1^	1.68 × 10^−1^	1.35 × 10^−1^	1.05 × 10^−1^	7.93 × 10^−2^
Benzyl Acetate	8.13	7.17	6.51	5.68	4.55	3.55	2.67
Benzyl Alcohol	1.01 × 10^2^	8.92 × 10^1^	8.10 × 10^1^	7.06 × 10^1^	5.66 × 10^1^	4.42 × 10^1^	3.33 × 10^1^
D5	9.06	7.99	7.26	6.33	5.08	3.96	2.98
Dimethicone 1	2.88	2.54	2.31	2.01	1.61	1.26	9.48 × 10^−1^
Dimethicone 2	3.87 × 10^3^	3.41 × 10^3^	3.10 × 10^3^	2.70 × 10^3^	2.17 × 10^3^	1.69 × 10^3^	1.27 × 10^3^
Ethanol	4.54 × 10^2^	4.00 × 10^2^	3.64 × 10^2^	3.17 × 10^2^	2.54 × 10^2^	1.98 × 10^2^	1.49 × 10^2^
Ethyl Acetate	1.83 × 10^−1^	1.62 × 10^−1^	1.47 × 10^−1^	1.28 × 10^−1^	1.03 × 10^−1^	8.01 × 10^−2^	6.03 × 10^−2^
Ethylbenzene	7.18 × 10^−1^	6.33 × 10^−1^	5.75 × 10^−1^	5.01 × 10^−1^	4.02 × 10^−1^	3.14 × 10^−1^	2.36 × 10^−1^
Hexyl Acetate	5.64	4.98	4.52	3.94	3.16	2.47	1.86
Limonene	4.12	3.63	3.30	2.87	2.31	1.80	1.35
2-Propanol	4.39 × 10^1^	3.88 × 10^1^	3.52 × 10^1^	3.07 × 10^1^	2.46 × 10^1^	1.92 × 10^1^	1.45 × 10^1^
t-Butyl Alcohol	1.33 × 10^1^	1.18 × 10^1^	1.07 × 10^1^	9.32	7.48	5.84	4.39
Toluene	4.73	4.17	3.79	3.30	2.65	2.07	1.56
Trimethylsilanol	1.94	1.71	1.56	1.36	1.09	8.50 × 10^−1^	6.40 × 10^−1^
Total	4.52 × 10^3^	3.98 × 10^3^	3.62 × 10^3^	3.15 × 10^3^	2.53 × 10^3^	1.97 × 10^3^	1.49 × 10^3^

**Table 4 ijerph-20-05944-t004:** Estimated VOC emission dose from standard head, neck, and face application of sunscreen based on age and sex (Table 3) calculated from mean emission factors in Figure 5 (Appendix A).

Species	Head, Neck, and Face Application Based on Age and Sex (mg)						
	Men (16+ Years)	Women (16+ Years)	Child (13–15 Years)	Child (11–12 Years)	Child (8–10 Years)	Child (5–7 Years)	Child (2–4 Years)
Benzene	2.19 × 10^−2^	1.93 × 10^−2^	1.75 × 10^−2^	1.52 × 10^−2^	1.23 × 10^−2^	9.56 × 10^−3^	7.20 × 10^−3^
Benzyl Acetate	7.39 × 10^−1^	6.51 × 10^−1^	5.92 × 10^−1^	5.15 × 10^−1^	4.14 × 10^−1^	3.23 × 10^−1^	2.43 × 10^−1^
Benzyl Alcohol	9.20	8.10	7.37	6.40	5.16	4.01	3.02
D5	8.24 × 10^−1^	7.26 × 10^−1^	6.60 × 10^−1^	5.74 × 10^−1^	4.62 × 10^−1^	3.60 × 10^−1^	2.71 × 10^−1^
Dimethicone 1	2.62 × 10^−1^	2.31 × 10^−1^	2.10 × 10^−1^	1.82 × 10^−1^	1.47 × 10^−1^	1.14 × 10^−1^	8.61 × 10^−2^
Dimethicone 2	3.52 × 10^2^	3.10 × 10^2^	2.82 × 10^2^	2.45 × 10^2^	1.97 × 10^2^	1.53 × 10^2^	1.16 × 10^2^
Ethanol	4.13 × 10^1^	3.64 × 10^1^	3.31 × 10^1^	2.87 × 10^1^	2.32 × 10^1^	1.80 × 10^1^	1.36 × 10^1^
Ethyl Acetate	1.67 × 10^−2^	1.47 × 10^−2^	1.34 × 10^−2^	1.16 × 10^−2^	9.35 × 10^−3^	7.27 × 10^−3^	5.48 × 10^−3^
Ethylbenzene	6.53 × 10^−2^	5.75 × 10^−2^	5.23 × 10^−2^	4.54 × 10^−2^	3.66 × 10^−2^	2.85 × 10^−2^	2.15 × 10^−2^
Hexyl Acetate	5.13 × 10^−1^	4.52 × 10^−1^	4.11 × 10^−1^	3.57 × 10^−1^	2.88 × 10^−1^	2.24 × 10^−1^	1.69 × 10^−1^
Limonene	3.74 × 10^−1^	3.30 × 10^−1^	3.00 × 10^−1^	2.61 × 10^−1^	2.10 × 10^−1^	1.63 × 10^−1^	1.23 × 10^−1^
2-Propanol	4.00	3.52	3.20	2.78	2.24	1.74	1.31
t-Butyl Alcohol	1.21	1.07	9.73 × 10^−1^	8.45 × 10^−1^	6.81 × 10^−1^	5.30 × 10^−1^	3.99 × 10^−1^
Toluene	4.30 × 10^−1^	3.79 × 10^−1^	3.45 × 10^−1^	2.99 × 10^−1^	2.41 × 10^−1^	1.88 × 10^−1^	1.41 × 10^−1^
Trimethylsilanol	1.77 × 10^−1^	1.56 × 10^−1^	1.42 × 10^−1^	1.23 × 10^−1^	9.92 × 10^−2^	7.72 × 10^−2^	5.81× 10^−2^
Total	4.11 × 10^2^	3.62 × 10^2^	3.29 × 10^2^	2.86 × 10^2^	2.3 × 10^2^	1.79 × 10^2^	1.35 × 10^2^

## Data Availability

The data presented in this study are available in Appendix A. Further data available on request.

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
