# Peer review of "Gas Phase Emissions of Volatile Organic Compounds Arising from the Application of Sunscreens"

_ijerph, 2023, doi:10.3390/ijerph20115944_

Round 1
Reviewer 1 Report
The subject of this paper is quite interesting. In this study, speciation of VOCs and gas phase emission rates of VOCs from a robust range of sunscreen products were evaluated, contributing to improve the understanding of the potential risk from sunscreen products usage in terms of VOCs exposure. Introduction is well organized and scientifically supported with adequate citations; key research questions were outlined. Methods and data analisys are clearly described, comprising typical procedures. Data are properly interpreted and results are discussed with proper citations. Conclusions are meaningful but limitations/suggestions for study improvements clould be explored.
Specific comments/questions
Lines 141-142: "The emissions estimates were made following the methodology reported previously in Yeoman et al. (2020)": I suggest to briefly describe this methodology, highlighting the most important aspects and assumptions here not reported. I believe that It it rather important to let the reader know why stailess steel gas-tight sample vessel was thermostatted at 35 ºC and the potential implications of this assumption on results.
Lines 142-143: apparently that information is not reported in Table 2.
Table 3: The description mentions “....age an sex (Table 3)“, I believe you wanted to refer to Table 2.
Figure 5: In the description, include the reference temperature of the estimated emission factors.
Conclusions: If there is any important suggestion for future studies, please report.
Author Response
The subject of this paper is quite interesting. In this study, speciation of VOCs and gas phase emission rates of VOCs from a robust range of sunscreen products were evaluated, contributing to improve the understanding of the potential risk from sunscreen products usage in terms of VOCs exposure. Introduction is well organized and scientifically supported with adequate citations; key research questions were outlined. Methods and data analisys are clearly described, comprising typical procedures. Data are properly interpreted and results are discussed with proper citations. Conclusions are meaningful but limitations/suggestions for study improvements clould be explored.
We have included the following in our conclusion further discussion of the study's limitations and suggestions for improvement:
With regards to the contaminant species:
"It should be noted that as the sample size for this investigation was small, the presence of contaminant species needs to be explored further in a separate and larger study."
Regarding inhalation exposure:
"These conclusions could be further consolidated by replicating real-life use and directly measuring inhalation using an inhalation replica"
Specific comments/questions
Lines 141-142: "The emissions estimates were made following the methodology reported previously in Yeoman et al. (2020)": I suggest to briefly describe this methodology, highlighting the most important aspects and assumptions here not reported. I believe that It it rather important to let the reader know why stailess steel gas-tight sample vessel was thermostatted at 35 ºC and the potential implications of this assumption on results.
Most of the methodology used in Yeoman et al 2020 has been described in the paragraph following (starting at line 128). However, we have added detail for clarity. The sentence now reads:
"The emissions estimates were made following the methodology reported previously in Yeoman et al. (2020)[19], with the SIFT-MS instrument sampling from the headspace of a gas-tight sample vessel where the sample is placed. Targeted VOCs and the m/z values used to identify them are listed in SI Table 2".
The following paragraph now reads:
"Approximately 2 mg of each product was weighed into a plastic vial cap and placed into a 50 mL stainless steel gas-tight sample vessel, which in turn was thermostatted at 35℃ for 1 hour to simulate skin temperature. Driving off VOCs at a higher temperature would not be representative of emissions during normal product use."
Lines 142-143: apparently that information is not reported in Table 2.
We refer here to the supplementary information (SI) Table 2.
Table 3: The description mentions “....age an sex (Table 3)“, I believe you wanted to refer to Table 2.
Corrected to Table 2
Figure 5: In the description, include the reference temperature of the estimated emission factors.
Have included `thermostatted at 35℃` so the caption now reads “Emission factors for 15 of the most commonly found VOCs in 26 different sunscreens thermostatted at 35℃ showing max, min and mean emissions . Excluded from the statistics are products where QTOF results showed a particular VOC was not present in the original formulation.”
Conclusions: If there is any important suggestion for future studies, please report.
Additions to the conclusion, and suggestions of future studies, have been outlined in the point above.
Reviewer 2 Report
The paper is interesting for conscious choices of the customers interested in this products.
It would be interesting to report the potential effects of exposure to voc for a well selected period of time considering the emission factors in fig5. In my opinion, the choice of the period of time should be done considering the habits of average customers.
Author Response
The paper is interesting for conscious choices of the customers interested in this products.
It would be interesting to report the potential effects of exposure to voc for a well selected period of time considering the emission factors in fig5. In my opinion, the choice of the period of time should be done considering the habits of average customers.
As this is a leave-on product VOCs will be completely off-gassed and exposure time will be the same for all users, unless the product is removed (as this is contrary to the end-use this would be unlikely). An applicant using more product, or using the product more frequently, will be exposed to a higher dose. This will scale linearly, as shown in Tables 3 and 4.
Reviewer 3 Report
The authors of the article entitled “Gas Phase Emissions of Volatile Organic Compounds Arising from the Application of Sunscreens" presented an evaluation of the variability and speciation of VOCs emitted from a range of sunscreen products, with the identification of potential trace contaminant aromatic VOCs in sunscreen products. After a thorough reading of the manuscript, there were some minor oversights that should be corrected in order for the article to fully meet the editorial requirements of the Journal.
|
Part of the manuscript |
Comment |
|
Discussion |
The authors should compare achieved results with the other authors. |
|
Conclusions |
Conclusions require more detail and specific research. |
|
Additional remarks |
In the text authors should correct the citations according to the MDPI standard. Throughout the manuscript, citations are misused by the authors, names should not be given and consecutive numbers of publications should be in square brackets. The font used by the authors in all figures should be the same as in the entire manuscript (Palatino Linotype). Authors should check the parentheses throughout the article. The authors should correct spaces between words throughout the text. |
The manuscript prepared by the authors is part of the aims and scope International Journal of Environmental Research and Public Health. After completing the above comments, the article will meet the editorial requirements and can be published in the Journal.
Author Response
The authors of the article entitled “Gas Phase Emissions of Volatile Organic Compounds Arising from the Application of Sunscreens" presented an evaluation of the variability and speciation of VOCs emitted from a range of sunscreen products, with the identification of potential trace contaminant aromatic VOCs in sunscreen products. After a thorough reading of the manuscript, there were some minor oversights that should be corrected in order for the article to fully meet the editorial requirements of the Journal.
|
Part of the manuscript |
Comment |
|
Discussion |
The authors should compare achieved results with the other authors. |
|
Conclusions |
Conclusions require more detail and specific research. |
|
Additional remarks |
In the text authors should correct the citations according to the MDPI standard. Throughout the manuscript, citations are misused by the authors, names should not be given and consecutive numbers of publications should be in square brackets. The font used by the authors in all figures should be the same as in the entire manuscript (Palatino Linotype). Authors should check the parentheses throughout the article. The authors should correct spaces between words throughout the text. |
The manuscript prepared by the authors is part of the aims and scope International Journal of Environmental Research and Public Health. After completing the above comments, the article will meet the editorial requirements and can be published in the Journal.
Discussion: As this is very novel work it is difficult to find appropriate studies with which to compare. We have discussed exposure dose and comparisons to that reported by Zhou et al. 2017 on page 17.
Conclusions: We have included the following in our conclusion further discussion of the study's limitations and suggestions for improvement:
With regards to the contaminant species:
"It should be noted that as the sample size for this investigation was small, the presence of contaminant species needs to be explored further in a separate and larger study."
Regarding inhalation exposure:
"These conclusions could be further consolidated by replicating real-life use and directly measuring inhalation using an inhalation replica"
Additional Remarks: We have edited the text to ensure names have been used in citations only where appropriate, for example when a method has been used or data/conclusions compared. This includes removing "particularly by Steinemann and others[(20,61,62)]" on page 14 and "Pawlowski et al. (2021)([43)] devised a method to evaluate all potential environmental impacts called the EcoSun Pass tool which score…" on page 11, which now reads "The EcoSun Pass tool [43] is a method used to evaluate all potential environmental impacts of UV blockers, scoring them on biodegradation, bioaccumulation, acute aquatic toxicity, chronic aquatic toxicity, sediment toxicity and chronic terrestrial toxicity."
Reference numbers have been placed in square brackets in line with the MDPI journal style guide.
Spaces corrected throughout text.